# Ultrastructural analysis in yeast reveals a meiosis-specific actin-containing nuclear bundle

Tomoko Takagi[1,2,3], Masako Osumi [3,4] & Akira Shinohara [1✉]

Actin polymerises to form filaments/cables for motility, transport, and the structural framework in a cell. Recent studies show that actin polymers are present not only in the cytoplasm but also in the nuclei of vertebrate cells. Here, we show, by electron microscopic observation with rapid freezing and high-pressure freezing, a unique bundled structure containing actin in the nuclei of budding yeast cells undergoing meiosis. The nuclear bundle during meiosis consists of multiple filaments with a rectangular lattice arrangement, often showing a feather-like appearance. The bundle was immunolabelled with an anti-actin antibody and was sensitive to an actin-depolymerising drug. Similar to cytoplasmic bundles, nuclear bundles are rarely seen in premeiotic cells and spores and are induced during meiotic prophase-I. The formation of the nuclear bundle is independent of DNA double-stranded breaks. We speculate that nuclear bundles containing actin play a role in nuclear events during meiotic prophase I.

[1] Institute for Protein Research, Osaka University, Suita, Osaka, Japan. [2] Laboratory of Electron Microscopy, Japan Women's University, Bunkyo, Tokyo, Japan. [3] Department of Chemical and Biological Sciences, Faculty of Science, Japan Women's University, Bunkyo, Tokyo, Japan. [4] NPO: Integrated Imaging Research Support, Chiyoda, Tokyo, Japan. ✉email: ashino@protein.osaka-u.ac.jp

In the cytoplasm, actin, as a cytoskeletal protein, polymerises to form a filament (F-actin) for various cellular functions, such as motility, division, phagocytosis, endocytosis, and membrane trafficking[1]. The dynamics of cytoplasmic actin filaments are highly regulated by various factors in different environments. Actin is also present in nuclei[2]. Actin monomer functions as a component of several chromatin-remodelling complexes for transcription and other nuclear events[3–6].

Recent studies have shown that polymerised forms of actin are present in the nuclei of various types of vertebrate and invertebrate cells[7–12]. Approximately forty years ago, Fukui and his colleagues identified actin bundles in *Dictyostelium* and HeLa cells upon treatment with dimethyl sulfoxide[13–15]. In *Xenopus* oocytes, nuclear actin forms a mesh of filaments, which is involved in the protection of nucleoli from gravity-induced aggregation[16]. In starfish oocytes, actin filaments promote the breakdown of the nuclear envelope and, by forming a mesh, the capture of chromosomes by spindles in cell division[17]. In mouse oocytes, actin filaments promote chromosome segregation during meiosis I and II[8]. Somatic mammalian cells transiently induce the formation of actin polymers in the nucleus in response to stress, serum starvation, heat shock, and DNA damage, such as DNA double-strand breaks (DSBs). Under serum starvation, F-actin participates in transcription by facilitating the activity of a transcriptional cofactor, MRTF (myocardin-related transcription factor)[18,19]. Nuclear F-actin also promotes the repair of DSBs in mammalian and fruit fly cells[7,10,20].

In budding and fission yeasts, actin is present in the cytoplasm in a polymerised form, such as rings, patches and cables[21–23], as well as the filasome, which is a less well-defined cytoplasmic amorphous structure containing F-actin[24]. In the budding yeast *Saccharomyces cerevisiae*, actin cables in the cytoplasm play a role in transport in mitotic budded cells. Previous electron microscopic (EM) analysis of the cytoplasm in fixed mitotic cells revealed a linear actin bundle/cable containing multiple actin filaments[25]. Actin polymers are also visualised by using a green fluorescent protein (GFP) fused with an actin-binding protein, Abp140, or by staining with an actin-specific peptide with a fluorescent dye such as phalloidin[26]. This staining confirmed the presence of filaments/cables and patches containing actin polymers in the yeast cytoplasm.

Actin polymers or cables are also present in the cytoplasm of meiotic yeast cells[27,28]. Actin cables are induced during meiotic prophase-I (meiotic G2 phase) and form a network that surrounds the nucleus. Cytoplasmic actin cables attached to protein ensembles in the nuclear envelope (NE) drive the motion of telomeres and thus chromosomes[27,29]. The fine structures of actin cables in meiotic cells are less defined. Previous EM observation of chemically fixed cells undergoing meiosis showed the synaptonemal complex (SC), a meiosis-specific chromosome structure, and nuclear microtubules from the spindle pole body (SPB), a yeast centrosome embedded in the NE[30–32]. However, little is known about the ultrastructures of cellular components inside meiotic yeast cells.

In this study, we analysed ultrastructures inside meiotic yeast cells by using freeze-substitution electron microscopy, which is suitable to observe near-native cellular structures[33]. Interestingly, we detected bundles inside nuclei in cells undergoing meiosis I but not in premeiotic cells or spores. The nuclear bundle is structurally similar to the cytoplasmic bundle induced in meiotic cells. The nuclear and cytoplasmic bundles are immunolabelled with anti-actin antibody and are sensitive to treatment with an actin-depolymerising drug. Meiosis-specific nuclear bundles consist of multiple filaments with a regular arrangement forming a three-dimensional structure. These results indicate that multiple bundles or bundle networks are formed inside the nuclei of cells

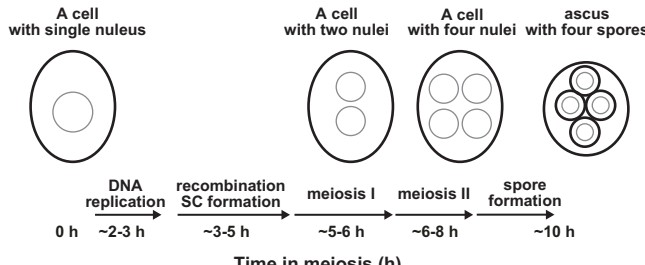

**Fig. 1 Schematic of yeast meiosis.** Times after the induction of meiosis of a single diploid yeast are shown with critical meiotic events. The nuclear membrane and cell wall are shown as grey and black lines, respectively.

undergoing unique chromosomal events in prophase I. Moreover, the nuclear bundle is seen in the *spo11* mutant, which is defective in the formation of meiotic DSBs for homologous recombination. The biological implications of nuclear bundles containing actin in meiotic cells are discussed.

## Results

**Electron microscopic observation of meiotic yeast cells**. To obtain more detailed information on ultrastructures of cellular structures and their spatial relationships with organelles inside meiotic yeast cells, we used transmission electron microscopy (TEM). Meiosis was induced by incubating yeast diploid cells in sporulation medium (SPM). Under this condition, wild-type cells carried out DNA replication and meiotic recombination from 2 h to 5 h after induction. At ~5 h, cells entered meiosis I, and by ~ 8 h, most of the cells finished meiosis II with a further developmental stage of sporulation (Fig. 1). Cells were quickly frozen, substituted with fixative and stained with osmium (freeze-substitution method)[33]. Thin sections of cells (50~60 nm) were observed under TEM (Fig. 2). In the freeze-substitution method, cellular organelles, including the nucleus, mitochondrion, and vacuoles in the cytoplasm filled with dense-stained ribosomes, were well preserved (Fig. 2). The nucleus was surrounded by double-layered nuclear membranes and contained electron-dense regions corresponding to the nucleolus (Fig. 2a, b, c). During meiosis prophase-I, i.e. 4 h after the induction of meiosis, a nucleus contacted a vacuole, forming a nuclear-vacuole junction (NVJ; Fig. 2f), as shown previously[34]. At 8 h, four prespore cells were formed inside the cells (Fig. 2j).

**Nuclear bundles in yeast meiosis**. In addition to known cellular structures/organelles, we detected a unique structure of bundles inside the nucleus of meiotic cells (Fig. 2c, d, e, f, h). This bundle is structurally different from microtubules in the nucleus emanating from the SPB (Fig. 3a, b). Sections of meiotic nuclei at late time points, such as 5 h, contained several bundles (Fig. 3), indicating that nuclear bundles are an abundant structure. Each bundle contained three to ten thin parallel filaments (Fig. 2e, h). The cross-section of the bundle showed an oval-like appearance of a single filament (Fig. 2e and Supplementary Fig. S1). The diameter of a single thin filament in nuclear bundles was ~7–8 nm (Fig. 4a; 8.3 ± 1.3 nm [mean ± S.D.; n = 17], 7.1 ± 1.1 nm [n = 8], 7.6 ± 1.0 nm [n = 13] for three independent sections; Supplementary Fig. 1).

In the bundle, filaments exhibited a regular rectangular/square arrangement (lattice) with repeated units of alternate single and double filaments (Fig. 2e; shown as red dots in Fig. 2e'). The average distance between adjacent filaments (Supplementary Fig. 1) was 8–15 nm (Fig. 4b; 15.2 ± 3.1 nm [n = 51], 8.3 ± 3.2 nm [n = 10], 12.2 ± 2.5 nm [n = 42]). The length of the bundles varied from 50 nm up to 1000 nm, near the diameter of

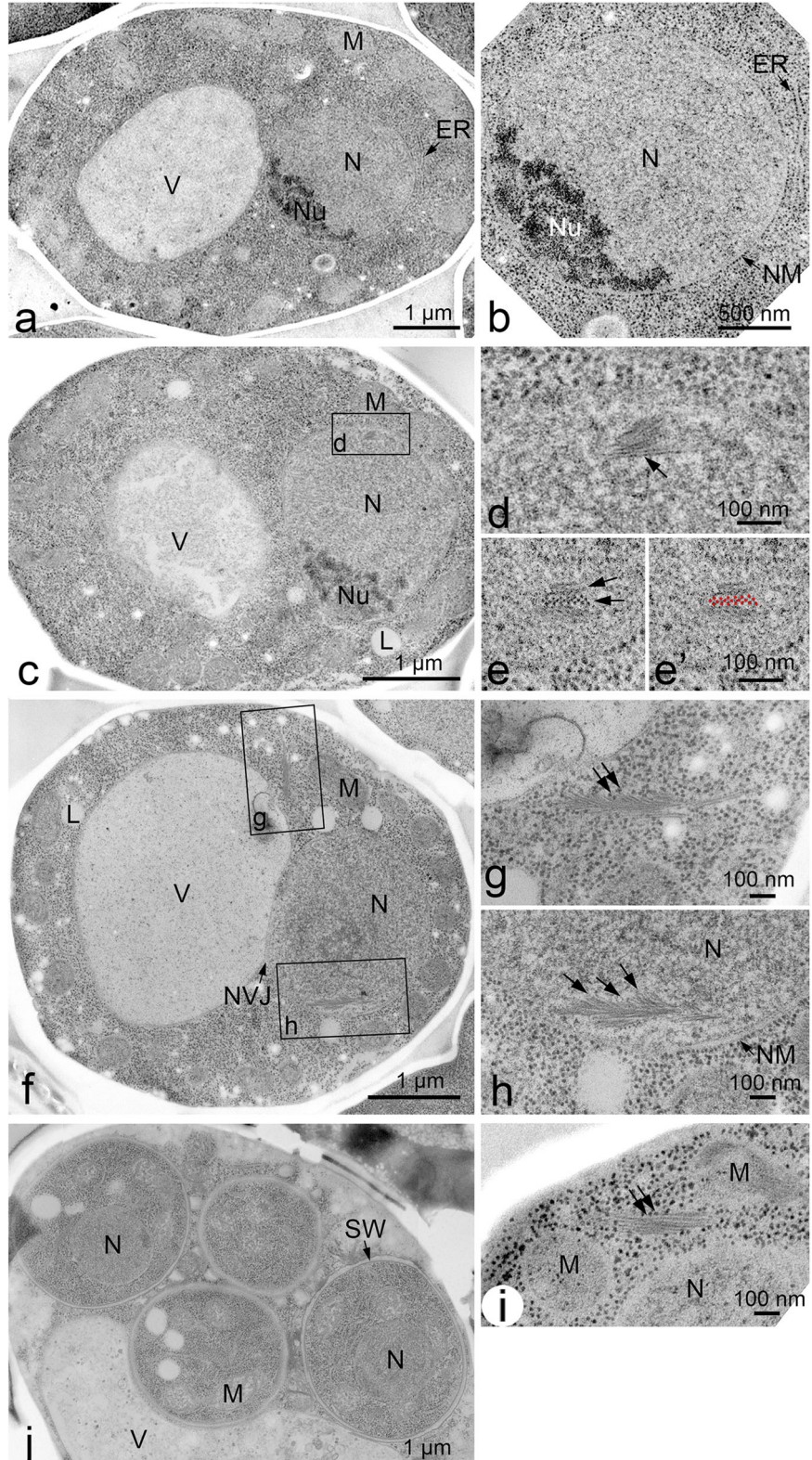

**Fig. 2 EM images of meiotic yeast cells. a, b** TEM images of a yeast diploid cell (MSY832/833) at 0 h. The specimens were prepared with freeze-fixation and sectioned. Magnified image is shown in (**b**). Bars indicate 1 μm and 500 nm in (**a**) and (**b**), respectively. **c–e**. TEM images of a yeast diploid cell at 2 h after incubation with SPM. Magnified views are shown in (**d**, **e**). **d** Vertical sections of bundles are shown by an arrow. Cross-sections of bundles are shown by arrows (**e**) and marked by red dots (**e'**). Bars indicate 1 μm (**c**) and 100 nm (**d**, **e**). **f–i**. TEM images of a yeast diploid cell at 4 h after incubation with SPM. A whole cell (**f**) and magnified views of the boxed region with bundles in the cytoplasm (**g**) and nucleus (**h**), where bundles of feather-like filaments (arrows) are seen. A bundle (arrow) is seen near the mitochondrion (**i**). Bars indicate 1 μm (**f**) and 100 nm (**g**, **h** and **i**). **j** TEM image of a yeast diploid cell at 8 h after incubation with SPM. Bar indicates 1 μm. ER endoplasmic reticulum, L lipid body, M mitochondrion, N nucleus, NM nuclear membrane (envelope), Nu nucleolus, NVJ nuclear-vacuole junction, SW spore wall, V vacuole.

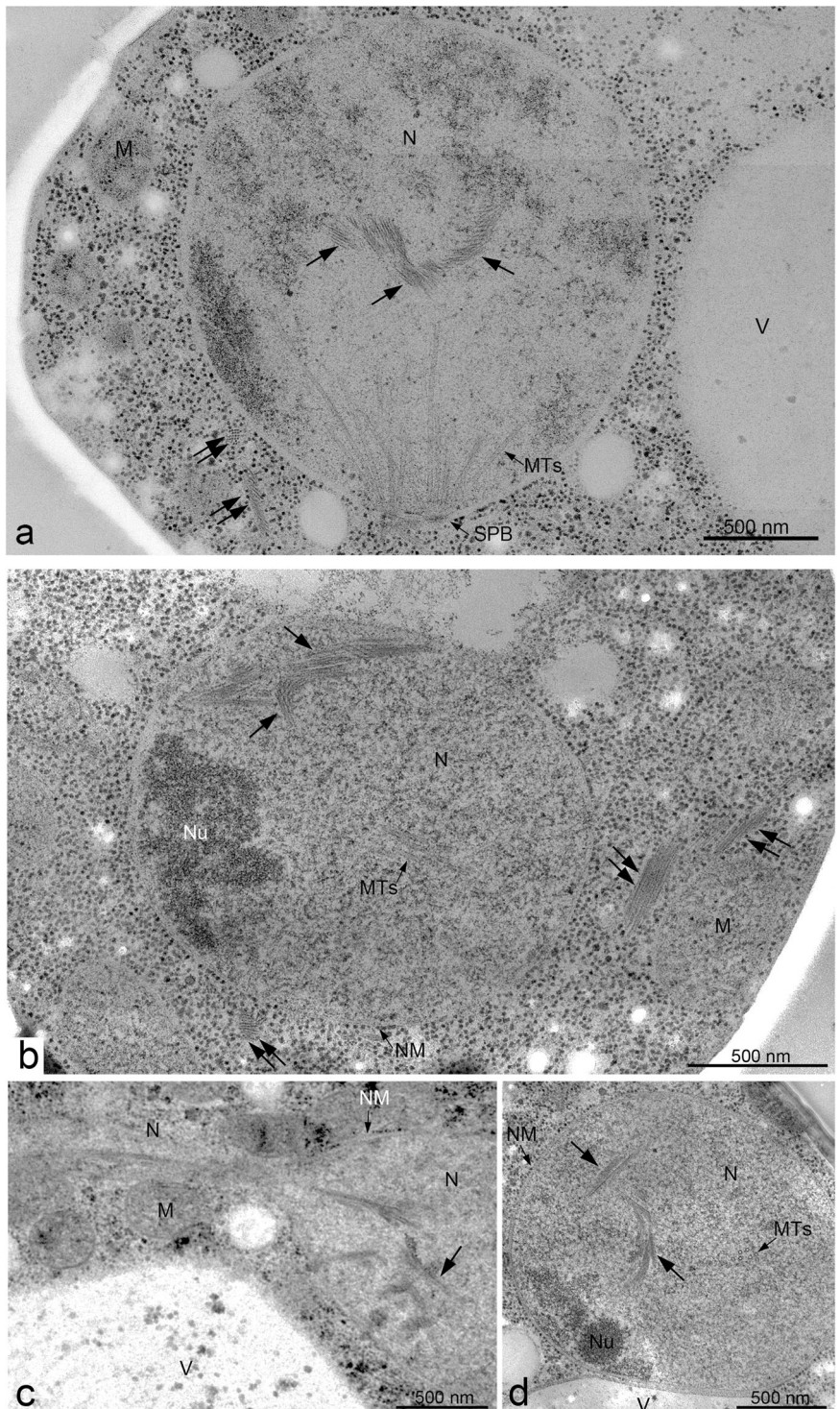

**Fig. 3 EM images of meiotic yeast cells during mid-prophase I.** TEM images of a yeast diploid cell (MSY832/833) after incubation with SPM for 4 h (**a**), 5 h (**b**) and 6 h (**c**, **d**). The specimens were prepared by cryofixation, freeze-substitution, and sectioning. **a**, **b**, and **d** Representative images of nuclear bundles with possible branches. **c** Microtubules and nuclear bundles are found in the stretched nucleus. Black arrows indicate nucleus bundles. Double black arrows indicate cytoplasmic bundles. Bars indicate 500 nm. M mitochondrion, MTs microtubules, N nucleus, NM nuclear membrane (envelope), Nu nucleolus, SPB spindle pole body, V vacuole.

the nucleus (Fig. 4c). The length of nuclear bundles in sections was 306 ± 223 nm [$n = 121$], median = 274 nm (Fig. 4c). In some cases, nuclear bundles spanned the whole nucleus (~1 μm, see Fig. 6-section 5). The longer bundles consisted of multiple short bundles rather than a single linear bundle (see Fig. 6-section 5).

**Spatial arrangement of nuclear bundles in meiosis**. To obtain more spatial information on nuclear bundles, we checked serial sections of a nucleus in yeast cells at 4 h in meiosis (Figs. 5, 6). To achieve in-depth freezing in specimens, we froze cells under high pressure[35]. With high-pressure freezing, specimens suitable for

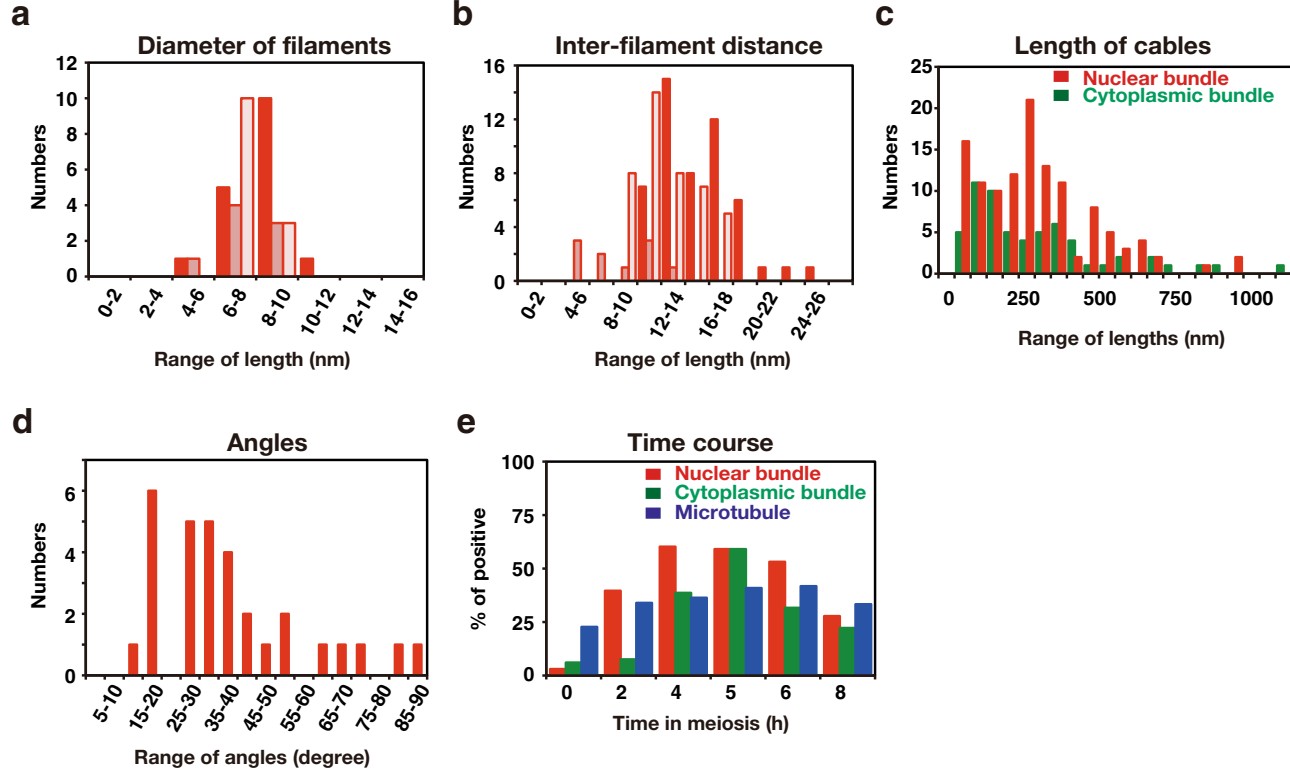

**Fig. 4 Quantification of nuclear and cytoplasmic bundles during meiosis. a** Distribution of filament diameters in nuclear bundles at 4 h in meiosis. The diameter of filaments in a cross-section of nuclear bundles was measured as shown in Supplementary Fig. 1c and ranked every 2 nm. The numerical rank in three independent time courses ($n = 17, 8, 13$) is shown in three different colours. **b** Distribution of the distance between two adjacent filaments in nuclei at 4 h in meiosis. The distance between two adjacent actin filaments in a cross-section was measured as shown in Supplementary Fig. 1f and ranked every 2 nm. The number of each rank in three independent images ($n = 51, 10, 42$) is shown in different colours. **c** Distribution of lengths of bundles with actin in nuclei and cytoplasm at 4 h in meiosis. The length of the bundles in a cross-section was measured and ranked every 50 nm. The number of bundles in nuclei (red) and cytoplasm (green) in a sum of three independent time courses is shown ($n = 60$ in cytoplasm, $n = 121$ in nucleus). **d** Distribution of angles of actin branches in nuclei at 4 h in meiosis. The angle between filaments and branched filaments was measured as shown in Supplementary Fig. 1g. The angles were ranked every 5 degrees. The number of each rank in a sum of three independent time courses ($n = 31$) is shown. **e** Kinetics of the formation of bundles in nuclei (red) and cytoplasm (green) as well as nuclear microtubules (blue) at different stages of meiosis. A number of sections containing bundles were counted, and the percentage of bundle-positive nucleus and cytoplasm sections as well as microtubule-positive nucleus sections are shown. Independent kinetic analysis is shown in Supplementary Fig. 2a.

sectioning were obtained. Importantly, we did not observe any change in subcellular structures, including bundles, in yeast cells prepared by either rapid freezing or high-pressure freezing (compare Fig. 2 with Figs. 5, 6). We often detected multiple nuclear bundles in different sections of 60 nm (Figs. 5, 6), indicating that the bundle is an abundant nuclear structure with a three-dimensional arrangement of bundles in the nucleus (Fig. 5). In some sections, a long bundle is observed that spans the entire nucleus (Fig. 6-sections 5 and 6), and the end of the bundles is likely to attach to the NE (arrows in Fig. 6-section 4–6).

Nuclear bundles often accommodate branch-like filaments or bundles, which look like "feathers" (Fig. 2g, h). At this resolution, some branched filaments appear to be attached to the lateral side of filaments in a main bundle, while other filaments are not attached. We measured the angle between the main bundles and the branch-like bundles/filaments (Fig. 4d, Supplementary Fig. 1). At 4 h, the angles between the main bundle and branch-like filaments were 25°–40° with subpeaks of 15°–20° (Fig. 4d; 38° ± 19° [$n = 31$], median = 34°). We noticed that branch-like filaments from a single bundle were oriented in the same direction, suggesting the directionality of the bundle.

**Nuclear bundles are similar to cytoplasmic bundles in meiosis.** We also detected bundles in the cytoplasm of meiotic yeast cells

(Fig. 2g, i). Structurally, the bundles in the cytoplasm were similar to those in nuclei (compare Fig. 2g, h). The diameter of a single thin filament in cytoplasmic bundles is approximately 7–8 nm (Supplementary Fig. 2a; 7.1 ± 1.0 nm [$n = 21$]), which is similar to that of a filament in nuclear bundles (Fig. 4a). The interfilament distance of the cytoplasmic bundles was 8–22 nm (Supplementary Fig. 2b). The length of cytoplasmic bundles in the sections (mean = 304 ± 179 nm [$n = 121$], median = 238 nm; Fig. 4c) was not different from that in the cytoplasm ($P = 0.64$, Mann–Whitney's $U$ test). These results indicated that nuclear and cytoplasmic bundles are structurally indistinguishable. While nuclear bundles form complicated 3D structures, cytoplasmic bundles form fewer such structures (Fig. 3b).

**Nuclear bundles by chemical fixation.** Yeast cells are surrounded by thick cell walls, which impede the penetration of staining reagents such as osmic tetroxide. We also stained spheroplasts of meiotic yeast cells with osmic acid after fixation with glutaraldehyde (without any freezing, Supplementary Fig. 3) and, in some cases, prepared serial sections for EM observation (Supplementary Fig. 4). With this procedure, the mitochondria showed high contrast (Supplementary Fig. 3). Membrane structures such as the nuclear membrane were partially deformed, possibly due to hypoosmotic conditions. Importantly, even under

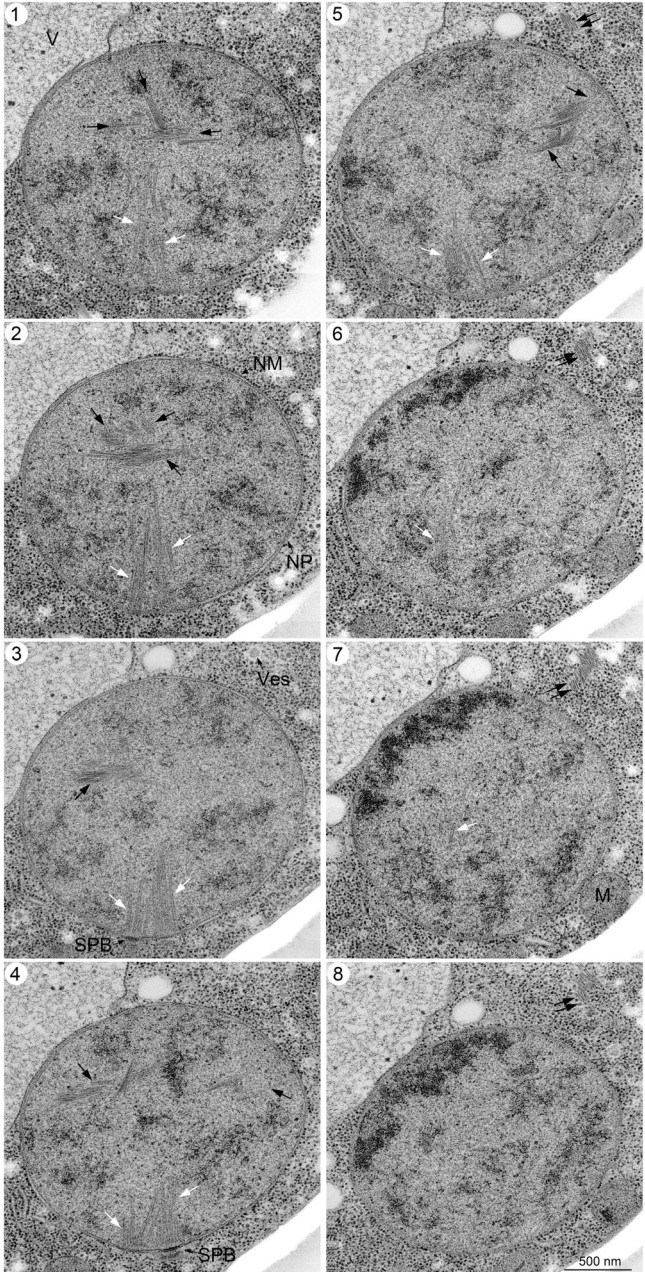

**Fig. 5 EM images of serial sections of yeast nuclei during mid-prophase I.** Specimens were prepared by high-pressure freezing as described in the Methods. Serial sectioned TEM images (sections 1–8; 60 nm section thickness) of a single cell at 4 h are shown. White and black arrows indicate spindle microtubules and nuclear bundles, respectively. Double black arrows indicate cytoplasmic bundles. Bar indicates 500 nm. M mitochondrion, NM nuclear membrane, NP nuclear pore, SPB spindle pole body, V vacuole, Ves vesicle.

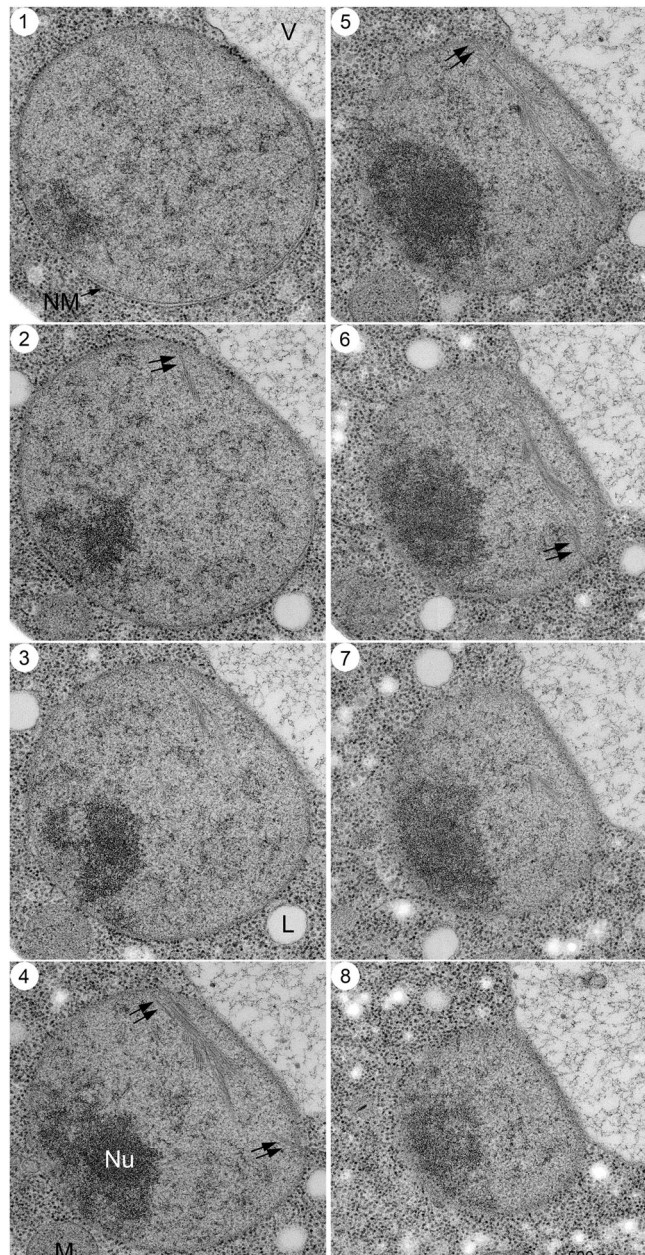

**Fig. 6 EM images of serial sections of yeast nuclei during mid-prophase I.** Serial sectioned TEM images (sections 1–8; 60 nm section thickness) of a cell at 4 h, which were fixed with high-pressure freezing, are shown. Black double arrows indicate nuclear bundles (sections 4–6), which are likely to attach to the nuclear envelope. Bar indicates 500 nm. M mitochondrion, NM nuclear membrane, Nu nucleolus, V vacuole.

this condition, we detected bundles in both the nucleus and cytoplasm of cells during meiotic prophase-I, and the structures and arrangement were similar to those obtained by the freeze-substitution method.

**Nuclear and cytoplasmic bundles are formed during prophase-I.** We checked the presence of nuclear bundles in different stages of meiosis (Fig. 4e). As a control, we measured nuclei containing microtubules. Nuclear microtubules were observed at 0 h, and sections positive for microtubules were increased slightly during

meiotic prophase-I. Nuclear bundles appeared earlier during meiosis than cytoplasmic bundles (Fig. 4e). At 0 h before the induction of meiosis, in which most cells were G1, we detected few bundles in the nuclei (Fig. 2a; 1.9% [0/10, 1/10 and 1/34], numbers of bundle-positive nucleus sections/numbers of total nucleus sections examined, three independent time courses) and cytoplasm (4.5% [1/10, 1/22 and 1/34] bundle-positive cytoplasm sections/total cytoplasm sections examined, three independent time courses). Few bundles were found in the four nuclei of prespore cells (Fig. 2j), suggesting that the bundle disassembled prior to the formation of prespore cells. Kinetic analysis revealed that nuclear bundles were seen at 2 h postinduction of meiosis

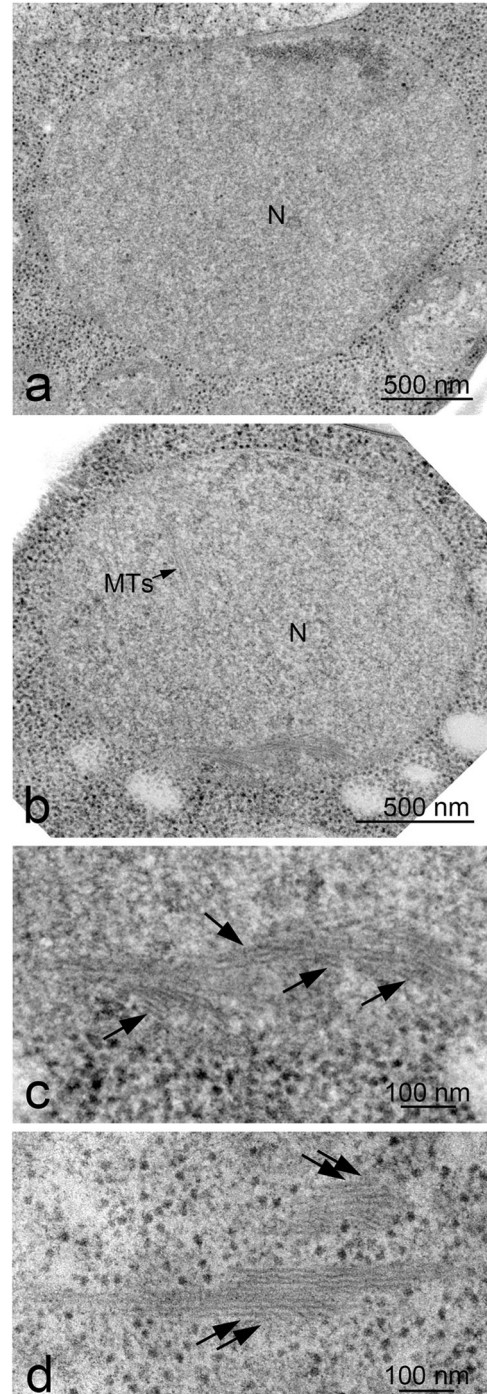

**Fig. 7 Nuclear bundles formed in the *spo11* mutant.** TEM images of a yeast *spo11-Y135F* diploid cell (HSY185/186) at 0 h (**a**) and 4 h (**b**). The specimens were prepared by freeze-fixation and sectioned. Magnified images are shown in (nucleus, **c** cytoplasm, **d**). Branch-like structure is shown in arrows. M mitochondrion, N nucleus, MTs microtubules. Bars indicate 500 nm (**a**, **b**) and 100 nm (**c**, **d**).

(Fig. 4e), when 38.9% (13/31, 8/22) of sectioned cells contained nuclear bundles, indicating that the formation of the structure is an early event of meiosis, e.g. during premeiotic S-phase. At 4 h, bundle-positive nuclei increased to ~60% frequency (1/12, 12/22, 15/22, 25/32; the numbers of bundle-positive nucleus sections/ numbers of total nucleus sections examined, four independent

time courses). Cells in late prophase-I at 5 h showed a peak in both bundle-positive nuclei and cytoplasm (Fig. 4e). After 6 h, both nuclear and cytoplasmic bundles decreased (Fig. 4e). At 6 h, when approximately half of the cells entered meiosis I, we detected the bundle in a bridge-like structure between two nuclei undergoing anaphase-I (Fig. 3c). This suggests that some nuclear bundles passed into the daughter nuclei during meiosis I.

**Nuclear bundles are formed in the absence of meiotic DSBs.** Since nuclear bundles are induced during meiotic prophase I, we analysed the relationship of the bundle with other prophase I events, such as meiotic recombination and SC assembly, both of which depend on the formation of meiotic DSBs by Spo11. We examined bundle formation in the *spo11-Y135F* mutant, which is defective in meiotic DSB formation[36,37]. At 0 h, there were few bundles in both the cytoplasm and nucleus in the mutant (Fig. 7a). As in wild-type cells, at 4 h, bundles were found in both the cytoplasm and nucleus in the *spo11-Y135F* mutant cells (41.2%, Fig. 7b). The kinetics of nuclear and cytoplasmic bundles (Fig. 7c, d) in the *spo11* mutant were similar to those in the wild type (Supplementary Fig. 5). These results indicate that the formation of nuclear bundles and cytoplasmic bundles are independent of meiotic DSBs induced by Spo11 and thus of meiotic recombination and SC formation.

**Nuclear and cytoplasmic bundles contain actin.** Previous EM analyses of mitotic cells of both budding and fission yeasts have shown three distinct actin subcellular structures: rings, cables, and patches as well as less-defined filasomes[21–24]. As a less-defined actin-related structure, the filasome is a cytoplasmic structure of less electron-dense areas with a vesicle in the centre (Supplementary Fig. 6) and is an actin-containing membrane-less subcellular structure in the cytoplasm originally found in fission yeast[24].

We initially found that the EM images of meiotic nuclear and cytoplasmic bundles are similar to those of actin cables in yeasts[21–25]. In addition, live visualisation of cytoplasmic actin polymers in meiotic prophase I with Abp140-GFP, which marks actin cables[27,28], indicated the presence of multiple actin cables located predominantly in the cytoplasm. It is known that the treatment of meiotic cells with an actin-depolymerising chemical, latrunculin B (LatB), greatly reduces cytoplasmic actin cables[27,28]. To confirm that the bundles in nuclei as well as in cytoplasm are indeed dependent on actin polymerisation, we treated 4-h meiotic yeast cells with LatB for 1 h and examined the bundles in cells under EM. As shown in Fig. 8a, b, the number of bundles (a section positive for bundles) was mildly reduced in nuclei after 1 h-treatment with LatB (from 60.2% [1/12, 12/22, 15/22, 25/32] at 4 h for positive nucleus to 25.3% [16/58, 4/21]; $P < 0.001$, Fisher's exact test). On the other hand, the cytoplasmic bundle is largely reduced by the LatB treatment (from 38.6% [34/88] to 2.5% [2/79]; $P < 0.001$). This indicates that some nuclear bundles are sensitive to LatB, supporting the idea that the formation of bundles in both the nucleus and cytoplasm requires actin polymerisation.

Previously, immuno-EM confirmed the presence of actin in the bundles in the cytoplasm of yeasts[38]. We performed immunogold labelling of sections of chemically fixed cells using an anti-actin antibody, which provides a less clear image than conventional EM (Fig. 8c, d and Supplementary Fig. 7, 8). Although not extensive, we found clustered gold labels on bundles in the nucleus and cytoplasm (Fig. 8c, d). We found 19/27 bundles (among 34 sections in two independent experiments) are positive for at least one gold particle. In the same sections, a ratio of cytoplasmic

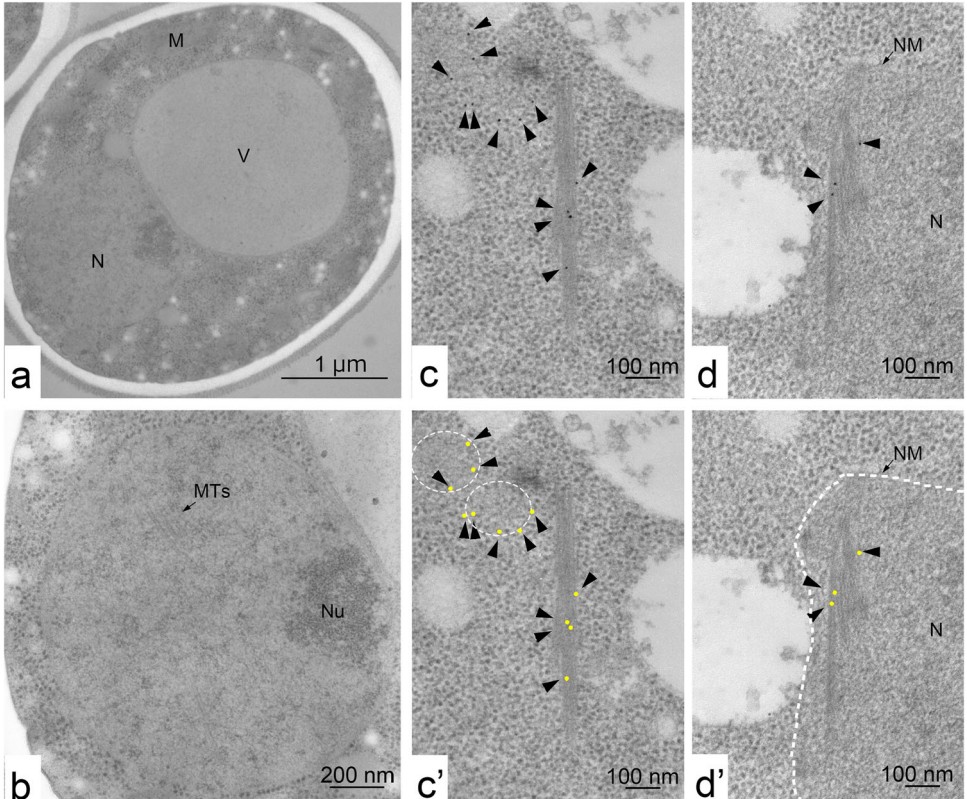

**Fig. 8 Nuclear bundles contain actin, and actin formation depends on actin polymerisation. a**, **b** TEM image of a latrunculin B-treated cell (**a**) and nucleus (**b**). Cells (MSY832/833) were incubated with SPM for 4 h and treated with latrunculin B for 1 h. The specimens were prepared by rapid freezing with fixation and sectioning. Bars indicate 1 μm (**a**) and 200 nm (**b**). M mitochondrion, N nucleus, NM nuclear membrane, Nu nucleolus, MTs microtubules, V vacuole. **c**, **d** Immunogold labelling using an anti-actin antibody (N350) was carried out as described in the Methods. Representative images of bundles in the cytoplasm (**c**) and nucleus (**d**) in a cell at 4 h using chemically fixed are shown. The positions of gold particles are shown in arrowheads. Yellow dotted circles are actin patches or filasomes (**c′**). The area surrounded by the white dotted line shows a nucleus (**d′**). The positions of gold particles are marked by yellow dots (**c′** and **d′**). Bars indicate 100 nm.

bundles positive to the gold is 6/7. In the nucleus, in addition to the bundle, we often detected particles in densely stained areas containing filament-like structures, which might be bundles. Importantly, nuclear bundles in freezing-fixation specimens showed more gold particles than other nuclear areas (Supplementary Fig. 7b). These results suggest that both nuclear and cytoplasmic bundles contain actin. We also detected gold particles in less electron-dense areas in the cytoplasm without ribosomes, which might correspond to the filasome (Fig. 8c).

## Discussion

In this study, by using TEM with rapid freezing-fixation, we found nuclear bundles containing actin in budding yeast cells undergoing the physiological programme of meiosis. We could also detect bundles in the cytoplasm, which are also induced during prophase I. Since we used rapid freezing to preserve the structures inside cells, it is unlikely that the bundle is an artefact produced by specimen preparation, which might be induced by external stress and/or staining. Moreover, we also detected bundles in nuclei fixed with chemicals without freezing (Supplementary Figs. 3, 4). We also noted that cryo-electron microscope tomography showed a nuclear bundle in meiotic yeast nuclei referred to as the "meiotic triple helix"[39], which is similar to the nuclear bundle described in this paper.

The nuclear bundles formed during meiosis appear to have a unique ultrastructure: multiple bundles accommodate three-dimensional arrangement, possibly through lateral interaction among the bundles and/or branch-like configuration. Nuclear

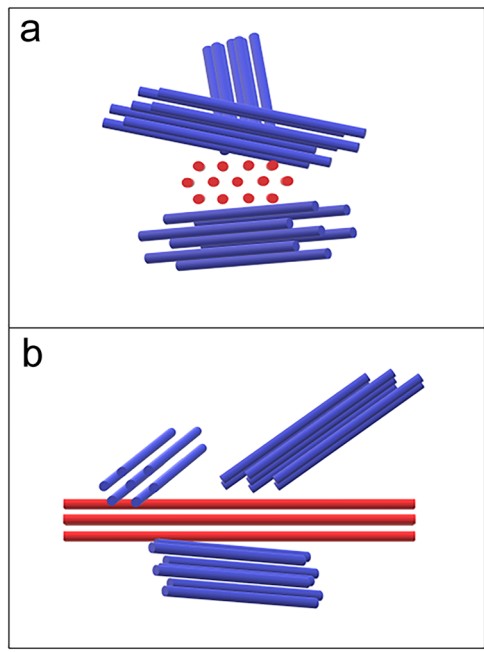

**Fig. 9 A model of nuclear bundles.** A schematic representation of bundles (red) with multiple branch-like filaments (blue) is shown. Schematic presentation of a cross-section of a nuclear bundle (**a**); arrangement of filaments in the bundle (**b**).

bundles seem to contain a unique arrangement of filaments, in which alternate patterns of 1 and 2 filaments are observed in a cross section of the bundles (red in Fig. 9a). This alternate pattern provides a rectangular/square arrangement of filaments in a single bundle. The interfilament distance is mainly 10–15 nm. Although we sometimes observed a branch-like structure of the bundle (Fig. 9b), we found no solid evidence on the branching of the filaments at the current EM resolution.

Nuclear bundles elongate up to 1 μm (Fig. 4c and 6). This long bundle consists of several bundles in an overlapping linear array rather than a single bundle, suggesting a self-assembly property of the bundle. We speculate that nuclear bundles are assembled into a long bundle with lateral attachment (or branching). Several bundles are present in a single nucleus of meiotic cells (Fig. 5, 6), indicating that nuclear bundles are abundant with a three-dimensional arrangement during late prophase I. We often see the bundle near the end of nuclear microtubules (Fig. 3a and 5–2). If abundant, how these bundles are packed in the context of nucleoplasm with meiotic chromosome structures such as SCs remains to be determined. Unfortunately, we could not efficiently detect SCs in our cryosections, although SCs could be detected by the heavy metals-staining of fixed meiotic cells[30–32].

Nuclear bundles were induced in very early meiotic prophase-I, such as 2 h after the induction of meiosis, and were present at least by meiosis I nuclear division. The bundles were abundant not only in nuclei but also in the cytoplasm, particularly during late prophase-I (Fig. 3b). On the other hand, we rarely observed nuclear or cytoplasmic bundles in G1 diploid cells (premeiotic) or spores (Fig. 2a, j). During meiosis, the formation of nuclear bundles starts at 2 h after the induction of meiosis, prior to DSB formation, which begins at ~3 h, indicating that nuclear bundle formation is not associated with DSB formation during yeast meiosis. Indeed, bundles are formed in the absence of meiotic DSBs in the spo11 mutant.

Our results here suggest that meiotic nuclear bundles contain actin and are sensitive to LatB (Fig. 8). The bundle is also sensitive to the other actin-depolymerising drug, latrunculin A[39]. Are the filaments in the nuclear and cytoplasmic bundles described here considered so-called typical actin filaments with a double-helix structure? The diameter of the filament in nuclear and cytoplasmic bundles is ~7–8 nm at minimum, which roughly corresponds with the size of a single yeast actin filament reconstituted in vitro[40,41]. However, the angle of putative branching in nuclear bundles is quite different from the angle of branches of actin filaments mediated by the Arp2/3 complex, which has a value of 70°[42]. Previous staining analyses with phalloidin or imaging with actin probes, Abp140-GFP[43] or Life-Act[39], has not shown the presence of any actin polymers inside yeast meiotic nuclei. These results suggest that the nuclear bundle is a novel structure containing actin. Indeed, cryo-electron tomography showed that the nuclear bundle contains triple-helical filaments[39].

What is the role of nuclear bundles? Actin polymerisation in the cytoplasm is involved in chromosome motion during prophase-I in budding yeast[43]. Cytoplasmic actin cables promote meiotic chromosome motion through SUN-KASH protein ensembles in the NE[29], which is sensitive to LatB[27]. This suggests that nuclear bundles play a role in the movement of meiotic chromosomes. Alternatively, nuclear bundles may protect nuclear structures from external forces generated during meiosis, as seen in Xenopus oocytes[16], by providing a rigid structure that resists mechanical stress generated through meiotic chromosome movement. In this line, we found that nuclear bundles form three-dimensional structures through self-assembly inside meiotic nuclei. Further studies are necessary to reveal the nature and function of nuclear bundles containing actin in meiotic cells.

## Methods

**Strains and culture.** The S. cerevisiae SK1 diploid strains MSY832/833 (MATα/MATa, ho::LYS2/", lys2/", ura3/", leu2::hisG/", trp1::hisG/") and HSY185/186 (MATα/MATa, ho::LYS2/", lys2/", ura3/", leu2::hisG/", spo11-Y135F-KanMX6/") were used for the meiotic time course.

Yeast cell culture and time-course analyses of the events during meiosis and cell cycle progression were as follows[44,45]. Briefly, 1 ml of diploid yeast culture in YPAD (1% yeast extract, 2% Bacto peptone, 2% glucose) was cultured in 200 ml of SPS (0.5% yeast extract, 1% Bacto peptone, 0.17% yeast nitrogen base, 1% KCH$_3$COO, 1% potassium hydrogen phthalate) medium for 16 h. Cells were collected and, after washing with H$_2$O, resuspended in 200 ml of SPM medium (0.3% KCH$_3$COO, 0.02% raffinose) to start meiosis.

**Rapid freezing for transmission electron microscopy.** Specimens for freeze-substitution electron microscopy were prepared according to a previously described method[46,47], with slight modifications. Cells were harvested by centrifugation. The cell pellets were sandwiched between two copper disks (3 mm in diameter). Specimens were quickly frozen with liquid propane using a rapid freezing device (KF80; Leica, Vienna, Austria). Specimens were freeze-substituted in cold absolute acetone containing 2% osmium tetroxide (OsO$_4$) at −80 °C for 48–72 h and were then warmed gradually (at −40 °C for 4 h, at −20 °C for 2 h, at 4 °C for 2 h and at room temperature for 2 h), washed with absolute acetone and rinsed with QY-2. Substitution for embedding was infiltrated with a Quetol-812 mixture (10, 30, 50, 70, 80 and 90 and 100% pure resin). The specimens polymerised at 60 °C for 2 days.

**High-pressure freezing fixation for transmission electron microscopy.** The specimens for high-pressure freeze-substitution electron microscopy were prepared according to a previously described method[48] with slight modifications. The pelleted cells were pipetted into aluminium specimen carriers (Leica) and frozen in an HPM-010 high-pressure freezing machine (BAL-TEC, Liechtenstein). The cells were transferred to 2% OsO$_4$ in cold absolute acetone. Substitution fixation was carried out at −90 °C for over 80 h. After fixation, the specimens were warmed gradually (at −40 °C for 2 h, at −20 °C for 2 h, at 4 °C for 2 h and at room temperature for 1 h), washed with absolute acetone and then exchanged with 0.1% uranyl acetate in absolute acetone. After staining, specimens were washed with absolute acetone and rinsed with QY-2. Substitution and embedding were as described above.

**EM grid preparation and observation.** Specimens for morphological observation were sectioned using an ULTRACUT-S ultramicrotome (Reichert-Nissei, Tokyo, Japan). Ultrathin sections were cut with a thickness of 50–60 nm and mounted on copper grids. Specimens for immunostaining were mounted on nickel grids. Ultrathin sections were stained with 4% uranyl acetate for 12 min in the dark at room temperature, placed in a citrate mixture (Sigma–Aldrich) for 2 min at room temperature and washed. Then ltrathin sections were examined using a JEM-1200EXS at 80 kV or a JEM-1400 transmission electron microscope (JEOL, Tokyo, Japan) at 100 kV.

**Immunoelectron microscopy and immunolabelling.** For immunoelectron microscopy, the chemical fixation specimens shown below were etched with 1% H$_2$O$_2$ for 5 min at room temperature. Otherwise, cell pellets were sandwiched between two aluminium disks (diameter 3 mm). Specimens were quickly frozen with liquid propane. They were freeze-substituted in cold absolute acetone containing 0.01% OsO$_4$. In other cases, specimens were fixed with a high-pressure freezing machine, then freeze-substituted in cold absolute acetone containing 0.1% glutaraldehyde. Substitution took place at −80 °C for 48–72 h, and the cells were then warmed gradually (at −40 °C for 4 h, at −20 °C for 2 h and at 4 °C for 1 h) and washed with dehydrated ethanol at 4 °C. Next, the cells were washed twice with cold ethanol and substituted at 4 °C, and infiltration was performed in an LR white and ethanol mixture (10, 30, 50, 70, 80, 90, 100% pure resin). Specimens were embedded in pure resin and polymerised at 50 °C for 1–2 days or at −35 °C in UV for 2 days.

Immunolabelling was carried out by a previously described method[48] with a slight modification. The sections mounted on nickel grids were incubated with chromatographically purified goat IgG (Zymed Laboratories, Inc., San Francisco, USA) diluted in blocking buffer; 0.1% bovine serum albumin (Sigma Chemical Co., St. Louis, USA) in 50 mM TBS (137 mM NaCl, 2.7 mM KCl, 50 mM Tris-HCl; pH 7.5) to a 1/30 concentration at room temperature for 30 min. For the chemical fixation specimens, an anti-actin antibody (anti-chicken gizzard actin, mouse monoclonal N 350; Amersham, Arlington, Heights, IL, USA) diluted in blocking buffer to 1/25 concentration was applied to the sections and incubated for 1 h at room temperature. For cryo-fixation specimens, affinity-purified mouse anti-actin, clone C4 monoclonal antibodies (MAB1501R: Chemicon International Inc., Temecula, USA and MAB8172: Abnova corp., Taipei, Taiwan, R.O.C.) diluted in blocking buffer to 1/50–1/100 concentration were applied to the sections and incubated for 16 h at 4 °C. The sections were washed with blocking buffer. Goat antibodies to mouse IgG labelled with 10 nm colloidal gold (British BioCell

International, Cardiff, UK) diluted in blocking buffer to 1/40 concentration were applied to sections for 1 h at room temperature. Sections were washed with blocking buffer and running water, then stained with a 4% aqueous uranyl acetate and citrate mixture (Sigma Chemical Co., St. Louis, USA).

**Latrunculin B treatment**. After culture in SPM medium for 4 h, cells were treated with 30 μM latrunculin B (R&D Systems) dissolved in 0.1% DMSO at 30 °C for 1 h. After the post-treatment, cells were collected and fixed by the rapid freezing method mentioned above.

**Chemical fixation method for electron microscopy**. Cells for chemical fixation were cultured with shaking for 4 h in SPM containing 1 M sorbitol (SPMS) at 130 rpm and 30 °C, as described above. Specimens for chemical fixation were shaken-cultured for 4 h in SPM containing 1 M sorbitol (SPMS) at 130 rpm 30 °C, details described above. Cells were treated for 2 min with 0.2 M DTT in ZK buffer (25 mM Tris-HCl and 0.8 M KCl). Then, the cells were treated for 30 min with 0.5 mg ml$^{-1}$ Zymolyase 100 T (Seikagaku Co., Tokyo) in ZK buffer at room temperature. Cells were treated with 0.5% Triton X-100 dissolved in PEM (20 mM PIPES, 20 mM MgCl$_2$, 10 mM EGTA, pH 7.0), protease inhibitors, and 1 M sorbitol for permeabilization in the presence of 4.2 μM phalloidin (Sigma–Aldrich) for 1 min. These treatments were performed as described previously[21,49].

Specimens for TEM were also prepared as described previously[24,49–51]. Briefly, harvested cells were fixed with 2% glutaraldehyde (Electron Microscope Sciences) and 0.2% tannic acid (TAAB) at 4 °C for 1 h. After washing with PEM, cells were postfixed in 2% OsO$_4$ in PEM at 4 °C for 1 h. After washing out with distilled water, cells were embedded in 2% agarose and cut into blocks smaller than 1 mm$^3$. Then, the specimens were dehydrated with an ethanol series (50%, 70%, 80%, 90%, 95%, 99.5%, and superdehydrated ethanol). Specimens rinsed with QY-2. The concentration of the resin infiltrated with a Quetol-812 (Nisshin EM, Tokyo) mixture (10, 30, 50, 70, 80, 90 and 100% pure resin) was gradually increased.

**Image processing and measurement**. The diameter of filaments and the distance between filaments were measured using ImageJ Fiji[52]. Cropped regions of TEM images were converted into binary images, and noise was removed. Filaments >10 nm$^2$ in size and 0.5–1.0 in circularity were recognised with the analysed particle function. Minor axis lengths of each particle were regarded as diameters of filaments. Distances between filaments were calculated using the central coordinates of particles. The number of ribosomes in filasome and cytoplasm of same in area were counted visually.

**Statistics and reproducibility**. The statistical significance of the difference in the length of bundles between the nucleus and cytoplasm was analysed using the Mann–Whitney's $U$ test. The null hypothesis was that there exists no variation in the length between the nucleus and cytoplasm. The statistical significance of the difference in the presence of bundles with actin in cells with and without LatB treatment was analysed using Fisher's exact test. The null hypothesis was that LatB treatment did not affect the existence of bundles in cells. The two-sided $P$ value was shown. Sample sizes are indicated in the corresponding figure legend and/or the main text.

**Reporting summary**. Further information on research design is available in the Nature Research Reporting Summary linked to this article.

## Data availability

The authors affirm that all data necessary for confirming the conclusions of the article are present within the article and figures. Yeast strains are available upon request. The source data of Fig. 4 and Supplementary Figs. 2 and 5 are provided as Supplementary Data 1. EM images are deposited in the Figshare platform (https://doi.org/10.6084/m9.figshare.15049380) and available upon request.

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

## Acknowledgements
We thank Drs. N. Nagata, I. Mabuchi, M. Sameshima, and N. Morone for discussion and suggestions. We also thank members of the Laboratory of Electron Microscopy, Japan Women's University and Ms. Y. Osaki in NPO: Integrated Imaging Research Support. We thank Drs. S. Gasser, V. Hurst, K. Shimada, I. Mabuchi, T. Yasunaga, and J. Usukura for critical reading of the paper. This work was supported by JSPS KAKENHI Grant Number: 22125001, 22125002, 15H05973 and 16H04742 to A.S and by the Open Research Center of JWU established in private universities in Japan with the support of the Ministry of Education, Culture, Sports, Science and Technology to M.O.

## Author contributions
T.T., M.O. and A.S. conceived and designed the experiments. T.T. performed all the E.M. experiments. T.T. and A.S. analyzed the data. T.T., M.O. and A.S. prepared the paper.

## Competing interests
The authors declare no competing interests.
