## [Peer Review File · Communications Biology]

Reviewers' comments:

Reviewer #1 (Remarks to the Author):

The ms by Takagi, Osumi and Shinohara describes 'nuclear actin cables with multiple branches during yeast meiosis'. The actin occurrence in the nucleus and its nuclear function are very open questions. In this regard the work by Takagi and colleagues is interesting. They provide a study based on electron microscopy of budding yeast meiotic nuclei by freeze- substitution method and chemical fixation. Nuclear cables, similar to cytoplasmic cables are detected 4h after induction of meiosis. These cables can be immuno-labeled with anti-actin antibodies. Cables disappear in the presence of Latrunculin B an inhibitor of actin polymerization. Measures were performed on diameters, length, inter-filament distances and angles.

As stated above the effort to provide evidence for nuclear form of actin is clearly beneficial to the field. The impressive filamentous structures seen by the authors call attention and deserve to be carefully examined in order to ensure that the cables observed are real actin structures. In this respect I suggest to:

- perform TEM *Sac6/fimbrin* mutants. It would also interesting to perform in TEM viable actin mutants
- ensure that nuclear structures are preserved in the presence of Latrunculin A, the $p = 0,0247$ between treated and non treated LatA cells is modestly significant.
- carefully distinguish actin from microtubules cables by immunogold anti-tubulin labeling together with anti-actin antibodies. Measures between actin and microtubules will also help to decipher between these two structures.

Minor comments.

Evidence for nuclear actin in yeast could be more clearly cited. Kapor, Fabre, Gasser and more recently Prado's work should be cited.

Please resolution accuracy should be given

What defines filasomes? please clarify

A scheme on the different steps of yeast meiosis during time will help the reader

How the distances between filaments are measured? please explain

Reviewer #2 (Remarks to the Author):

In this study, Takagi et al. observe actin cables with "feather-like" appearance in the cytoplasm and inside of the nucleus of *Saccharomyces cerevisiae* cells undergoing meiosis. They use different electron microscopy techniques, including rapid freezing and chemical fixation, to demonstrate that these structures are not an artifact. Also, they use immuno-labeling with anti-actin antibody and actin-depolymerizing drug to show that these cables contain actin. They demonstrate these structures increase during meiotic prophase (4h-5h after induction of meiosis) and analyze different parameters as diameter of nuclear actin cables, average distance between adjacent filaments, length of cables and angle between the main actin cables and branched structures.

This study demonstrates the presence of nuclear actin in the nucleus of budding yeast, which was not previously explored. Other studies have described in different organisms the existence of nuclear actin in response to stress, but here, this study demonstrates the existence of nuclear actin under physiological conditions, such as during meiotic prophase. They describe extensively different parameters of nuclear actin cables, but only briefly suggest its possible functions. Experiments with an actin-fluorescent probe, such as Life-Act::GFP with a nuclear localization sequence, would allow the observation of nuclear actin in live cells and would help to address the role of actin nuclear.

Major comments:

1) The authors use immuno-labeling with anti-actin antibody and also experiments with an actin-depolymerizing drug to conclude that the observed nuclear cables actually contain actin. In the first experiment, I agree with their interpretation of the results, but the number of dots of actin in cytoplasm and nucleus is much lower than the number described in other studies for cytoplasmic cables (reference 32). Since there is no negative control, I would like to be sure that these spots are really actin and not a consequence of non-specific binding of the antibody. Perhaps, the use of LatB treatment combined with the immuno-gold labeling experiment may help to address this issue. Alternatively, mutants defective in actin filament assembly could also be tested. In the second experiment, I consider that it is necessary to explain better the paragraph (line 161-167) to understand the results of this experiment. What does [1/12, 12/22, 15/22...] mean? number of actin cables or nuclei with actin cables/total analyzed of section by experiment?

2) It is not clear how the authors calculated the average distance between adjacent filaments (p7, line 205, Figure 3d.)? Have you obtained this data as the mean of experiments (15.2; 8.3 and 12.2)? Is 15 nm right?

3) Since the nuclear actin cables seem to be more prominent during meiotic prophase I, it would be interesting to test whether their appearance depends on recombination/synapsis by analyzing a spo11 mutant

Minor points:

p6, line 162. Move "after the treatment with LatB" in line 162 to line 164.

p7, line 197. Add "e'" in Figure 1.

p8, line 230. Please indicate "Fig.7 section 5 and 6" instead of "Fig.7-5 and 7-6", for easier understanding. Similar in line 231.

p8, line 244. In Supplementary Fig.3, I do not observe actin cables attached to the inner nuclear membrane; please, indicate it with arrows similar to Figure 7.

p19 and p20, line 570 and 623 (Ref.36 and 53, respectively). Write the name of organisms in italics, please.

p21, line 655. Add: Bars indicate 100 nm.

p21, line 659 and p24 line 731. Some abbreviations, such as L and NM, are shown in figure legends, but not in figures.

p22, line 666. Change symbol "%" to percentage.

p22, line 676 and 677. Colors described in the figure legend are not corresponding with colors of graphics.

p24, line 727-732. Please, revise this figure. In (b) I do not see actin cables; if I am wrong, please indicate its localization with arrows. (c) is not a magnified image, revise line 730. Add "N" in Supplementary Figure 2 (b,d,e,g) to indicate nucleus. Bars indicate 200 nm in (b,d,e,g), not 100 nm.

Figure 3, panel "e". The graph title must be "Angles" instead of "Angels".

Reviewer #3 (Remarks to the Author):

The goal of the studies described in Takagi et al. (COMMSBIO-19-1264-T) was to investigate the existence of and describe the structure of actin cables in the cytoplasm and nuclei of budding yeast, undergoing meiosis. Yeast cells use actin-based transport along cables to direct polarized cell growth and to segregate organelles prior to cell division (Bretscher, A. 2003). The existence of actin cables in the nucleus may play a somewhat different role; as such, the study has potential.

The authors used rapid freezing and high-pressure freezing, freeze-substitution and chemical fixation methods to identify a unique polymerization form of actin, both in the nucleus and in the cytoplasm. To prove that they really saw actin cables, authors labelled them with anti-actin antibodies and visualized by nano-gold; additionally, some cells were subjected to the treatment with an actin-depolymerizing drug – latrunculin, before fixation. After that, no cables remained in the cells.

On the whole, the paper is not very convincing. Images of the high pressure frozen yeast cells are acceptable, but the interpretation is not good enough. Below, the Major Concerns are listed:

- 1) The title of the paper is incorrect. It may be used as a running title, but the authors need to find more appropriate name for the primary title.
- 2) The immuno-gold labelling showed some gold particles associated with a bunch of the cables in the cytoplasm (Fig. 2a), but some gold particles were also identified in the cytoplasm. The authors suggest that these might be filasomes; they mentioned that filasomes are a “novel actin-containing membrane-less sub-cellular structure”, but the paper mentioning filasomes was published more than 15 years ago, which is not novel enough, in my opinion. Additionally, those filasomes from 2003 do not look similar to the current Fig. 2A. I suggest that this is just non-specific binding.
- 3) Line 325 – The authors speculate that actin cables may form a three-dimensional structure in the meiotic nuclei, but they did not attempt to prove this, for example, using the tomography approach.
- 4) The TEM images in Fig. 5 do not give an impression that the discovered actin cables have a novel branch pattern, schematized in Fig. 8. The serial sections in Fig. 6-7 are too thick to accommodate a single actin cable. Why should the “branches” be attached to the main cable? Maybe they just cross the plane with the main cable? Again, this issue may be resolved by the tomography approach.
- 5) Line 294 - The authors declare that they find an alpha-actinin-like protein in the budding yeast, but this suggestion is based only on the distance between filaments in the bundle. No immunoblotting was done to prove this.

Minor issues:

Fig 3e - Angles should be angles

Responses to reviewers:

We do appreciate for constructive comments on our paper by three reviewers. We are trying to answer the comments shown in green in below as much as possible. Prior to mentioning point-by-point responses to reviewers' comments, we will discuss related issues and summarize our major changes in the revised form.

When our manuscript has been reviewed, a paper by Ma et al. entitled "Meiotic budding yeast assemble bundled triple helices but not ladders", which was originally entitled "Cryo-ET analysis of budding yeast synaptonemal complexes in situ" as an original submission in BioRxiv (<https://doi.org/10.1101/746982>).

After discussing with us, the authors recently revised their paper as described above. In this paper, the authors carried out cryo-electron tomography (Cryo-ET) of meiotic yeast cells. Importantly, the authors identified the bundles of filaments in a meiotic nucleus of budding yeast cells, which look almost identical to what we reported in our submission. The authors also reported that the bundles are sensitive to Latruncurin-A, an actin depolymerizing drug, and a high concentration of 1,6 hexanediol. We believed that the bundle in their paper is same to nuclear filaments/bundles as described in the paper. Importantly, the authors showed a filament in the bundle forms "triple helices", not double helices known for a conventional actin filament. These results clearly confirmed our results with additional information and very nicely complementary to ours.

In our revised version, we added the following major changes.

1. To know the relationship of the nuclear bundle structure with prominent events during meiotic prophase I, such as meiotic recombination and the synaptonemal complex (SC), we carried out the EM analysis of a mutant of the *SPO11* gene (Klapholz & Esposito, 1982), the *spo11-Y135F* mutant, which is deficient in the formation of DNA double-strand break (DSB) during meiosis, an initiation event for meiotic recombination. Importantly, the *spo11* mutant is also defective in SC formation. Our new result showed that the *spo11* mutant is proficient in the formation of nuclear bundles/cables (Figure 7). This supports our conclusion that the cable structure is unrelated to the recombination, thus DNA damage response. This analysis was requested by #2 reviewer. Ma et al. (preprint) also showed the same results using the *spo11* deletion mutant.
2. We repeated the immuno-gold labelling using anti-actin antibody and added several representative images of the labelling of bundles (Supplemental Figure 7). As shown, due to technical difficulty, the immuno-gold labelling on the cables is not extensive (not efficient) under the condition. However, we did see "multiple/clustered" gold dots on nuclear bundles compared to the other area of nucleoplasm. We also softened our wording to describe the nature of the bundles by not mentioning the "actin cable".

3. We reorganized all Results sections and rewrote it in more fair way in our interpretation on nuclear “actin cable” than a previous version, which may be over-interpretation on our data. Since we have not had “firm” evidences on that nuclear and cytoplasmic bundles/cables described here consist of a *bona fide* actin filament, we do not use “actin cable” in the text and rather describe them as “bundles containing actin”.

Keeping in mind on the preprint by Ma et al., we wrote our responses as shown below (our responses shown in green).

Reviewers' comments:

Reviewer #1 (Remarks to the Author):

The ms by Takagi, Osumi and Shinohara describes ‘nuclear actin cables with multiple branches during yeast meiosis’. The actin occurrence in the nucleus and its nuclear function are very open questions. In this regard the work by Takagi and colleagues is interesting. They provide a study based on electron microscopy of budding yeast meiotic nuclei by freeze- substitution method and chemical fixation. Nuclear cables, similar to cytoplasmic cables are detected 4h after induction of meiosis. These cables can be immuno-labeled with anti-actin antibodies. Cables disappear in the presence of Latrunculin B an inhibitor of actin polymerization. Measures were performed on diameters, length, inter-filament distances and angles.

As stated above the effort to provide evidence for nuclear form of actin is clearly beneficial to the field.

The impressive filamentous structures seen by the authors call attention and deserve to be carefully examined in order to ensure that the cables observed are real actin structures. In this respect I suggest to:

- perform TEM Sac6/fimbrin mutants. It would also interesting to perform in TEM viable actin mutants

>>As pointed by the reviewer, delineating the nature of nuclear bundles is very important, but we believe that is for future studies. Since, although the nuclear bundles/cables contain actin, it is likely that the filament in nuclear bundles is not a typical actin filament with double helix. Moreover, the SAC6 gene is essential for cell viability, we need to construct a conditional mutant. We created a meiosis-specific null mutant of the SAC6 by using the CLB2 promoter, whose activity is down-regulated during meiosis, but could not succeed in depletion of the protein in this mutant (since the protein is stable even after shutdown of the expression). Moreover, the pre-print paper described above showed a unique structure of the filament with triple helix. We think, we cannot apply our knowledge on “conventional” actin filament to bundles/cables described in the paper. We feel that we should focus on cytological characterization on the bundle/cable and/or the filament as a primary focus in the paper.

- ensure that nuclear structures are preserved in the presence of Latrunculin A, the $p = 0,0247$ between treated and non treated LatA cells is modestly significant.

>> We agree the P-value we provided is modestly significant (<0.05) in the effect of the treatment with Latrunculin B for “one hour” (to minimize chronic effects) on the frequencies of the presence of nuclear bundles/cables. As written in the text, the effect of the drug is not drastic (even previous study showed mild reduction of cytoplasmic actin cables detected by ABP140-GFP with lots of residual patches by the treatment; Koszul et al. 2009). Consistent with our data, the authors in the pre-print used Latrunculin A (Not Latrunculin B used by us) and found the treatment with LatA decreased the presence of nuclear bundles.

- carefully distinguish actin from microtubules cables by immunogold anti-tubulin labeling together with anti-actin antibodies. Measures between actin and microtubules will also help to decipher between these two structures.

>> Given low efficiency of immuno-gold labelling in the fixed section, double-labelling is not a good way (in addition, to find a section with both bundles and microtubules is hard). Rather, to address the comment, by repeating the experiments, we added some images containing both nuclear cables and microtubules (Fig S1), which clearly shows structural difference between microtubules and nuclear bundle. In addition to Fig.8, we added multiple images of gold-labelling of anti-actin antibody as Supplemental Fig. 7.

Minor comments.

Evidence for nuclear actin in yeast could be more clearly cited. Kapor, Fabre, Gasser and more recently Prado's work should be cited.

>> We cited papers in nuclear actin by Kapor, Fabre, and Gasser in addition to recent paper by Prado's group (Genetics) in the end of the first paragraph in page 3.

Please resolution accuracy should be given

>> We could not catch the meaning of “resolution accuracy”. We thought about the low resolution of images in the paper. We used more than 300 dpi in all original images in Figures, but probably the resolution of the images might have been decreased due to the conversion to PDF. Importantly, we deposited original images with high resolution.

What defines filasomes? please clarify

>> We explained about the filasome and added a representative image in Supplemental Fig. 6 and explained more in the text (page 8, last line to page 9, line 1-4). Filasome is a cytoplasmic structure containing a small vesicle in the center which is surrounded by area without ribosomes.

A scheme on the different steps of yeast meiosis during time will help the reader .

>> We explained different steps of budding yeast meiosis in page 5, first paragraph, and also added a scheme of yeast meiosis in terms of time of synchronous meiosis in a new Fig.1.

How the distances between filaments are measured? please explain

>> We added the sentences on how we measured distance between the filament; we measured a distance between center points of two filaments, which is described as a schematic presentation (see Supplemental Figure 1f).

Reviewer #2 (Remarks to the Author):

In this study, Takagi et al. observe actin cables with “feather-like” appearance in the cytoplasm and inside of the nucleus of *Saccharomyces cerevisiae* cells undergoing meiosis. They use different electron microscopy techniques, including rapid freezing and chemical fixation, to demonstrate that these structures are not an artifact. Also, they use immuno-labeling with anti-actin antibody and actin-depolymerizing drug to show that these cables contain actin. They demonstrate these structures increase during meiotic prophase (4h-5h after induction of meiosis) and analyze different parameters as diameter of nuclear actin cables, average distance between adjacent filaments, length of cables and angle between the main actin cables and branched structures.

This study demonstrates the presence of nuclear actin in the nucleus of budding yeast, which was not previously explored. Other studies have described in different organisms the existence of nuclear actin in response to stress, but here, this study demonstrates the existence of nuclear actin under physiological conditions, such as during meiotic prophase. They describe extensively different parameters of nuclear actin cables, but only briefly suggest its possible functions. Experiments with an actin-fluorescent probe, such as Life-Act::GFP with a nuclear localization sequence, would allow the observation of nuclear actin in live cells and would help to address the role of actin nuclear.

Major comments:

1)The authors use immuno-labeling with anti-actin antibody and also experiments with an actin-depolymerizing drug to conclude that the observed nuclear cables actually contain actin. In the first experiment, I agree with their interpretation of the results, but the number of dots of actin in cytoplasm and nucleus is much lower than the number described in other studies for cytoplasmic cables (reference 32). Since there is no negative control, I would like to be sure that these spot are really actin and not a consequence of non-specific binding of the antibody. Perhaps, the use of LatB treatment combined with the immuno-gold labeling experiment may help to address this issue. Alternative, mutants defective in actin filament assembly could also be tested. In the second experiment, I consider that it is necessary to explain better the paragraph (line 161-167) to understand the results of this experiment. What does [1/12, 12/22, 15/22...] mean? number of actin cables or nuclei with actin cables/total analyzed of section by experiment?

>> This reviewer suggested two nice experiments. The first one is immuno-gold

labelling in a yeast cell treated with LatB. However, as shown, the LatB treatment did not remove all of the bundles, which makes difficult to evaluate the results of such immuno-EM experiment which does not provide robust staining. For the second experiment, the reviewer suggested a mutant which is deficient in actin filament assembly. As written in response to #1 reviewer, as shown in the pre-print by Ma et al, the structure of the bundle could be an atypical actin filament with triple helixes if it is a *bona fide* actin filament. So, we think that conventional mutants defective in actin filament formation might not be applicable to dissect the nature of nuclear cables here. As described above, we remove a word of “actin cables/bundles” from the text to avoid misunderstanding and rather used “nuclear bundle/cable containing actin”. We think that this word reflects what we observed in our study.

2) It is not clear how the authors calculated the average distance between adjacent filaments (p7, line 205, Figure 3d.)? Have you obtained this data as the mean of experiments (15.2; 8.3 and 12.2)? Is 15 nm right?

>> As described in response to #1 reviewers, we added the sentences on how we measured distance between the filament by measuring a distance between center points of two filament, which is described as a schematic presentation (see Supplemental Figure 1f).

Thanks. We corrected to 8-15 nm.

3) Since the nuclear actin cables seem to be more prominent during meiotic prophase I, it would be interesting to test whether their appearance depends on recombination/synapsis by analyzing a *spo11* mutant

>> We added our EM analysis of the *spo11-Y135F* mutant (Fig. 7), which show normal formation of actin bundles in the meiotic nuclei of the mutant, in Figure 7. This supports the idea that actin bundles are independent of recombination and SC formation, which depends on Spo11-induced DNA double-strand breaks.

Minor points:

p6, line 162. Move “after the treatment with LatB” in line 162 to line 164.

>> We moved.

p7, line 197. Add “e” in Figure 1.

>> We added “e”.

p8, line 230. Please indicate “Fig.7 section 5 and 6” instead of “Fig.7-5 and 7-6”, for easier understanding. Similar in line 231.

>> We changed them as pointed out.

p8, line 244. In Supplementary Fig.3, I do not observe actin cables attached to the inner nuclear membrane; please, indicate it with arrows similar to Figure 7.

>> The review is right. Since we have not had any firm evidence on the attachment of the cable to the nuclear envelope (only few cases among more than 100 sections) and softened our wording (page 6, last sentence).

p19 and p20, line 570 and 623 (Ref.36 and 53, respectively). Write the name of organisms in italics, please.

>> We italicized the name of organisms.

p21, line 655. Add: Bars indicate 100 nm.

>> We added the bar.

p21, line 659 and p24 line 731. Some abbreviations, such as L and NM, are shown in figure legends, but not in figures.

>> "NM" is shown in Fig. 2b and 2b' (now Fig. 8b and 8b'). We removed "L" from Legends.

p22, line 666. Change symbol "%" to percentage.

>> We changed it.

p22, line 676 and 677. Colors described in the figure legend are not corresponding with colors of graphics.

>> We removed the explanation on colors since it uses only one color.

p24, line 727-732. Please, revise this figure. In (b) I do not see actin cables; if I am wrong, please indicate its localization with arrows. (c) is not a magnified image, revise line 730. Add "N" in Supplementary Figure 2 (b,d,e,g) to indicate nucleus. Bars indicate 200 nm in (b,d,e,g), not 100 nm.

>> Since Fig. S1(a) and (b) are images from yeast cells at 0 h (before meiosis). We added arrows to show nuclear or cytoplasmic cables in images and put "N" to indicate nuclei. We corrected bars to 200nm.

Figure 3, panel "e". The graph title must be "Angles" instead of "Angels".

>> We changed as pointed out.

Reviewer #3 (Remarks to the Author):

The goal of the studies described in Takagi et al. (COMMSBIO-19-1264-T) was to investigate the existence of and describe the structure of actin cables in the cytoplasm and nuclei of budding yeast, undergoing meiosis. Yeast cells use actin-based transport along cables to direct polarized cell growth and to segregate organelles prior to cell division (Bretscher, A. 2003). The existence of actin cables in the nucleus may play a somewhat different role; as such, the study has potential.

The authors used rapid freezing and high-pressure freezing, freeze-substitution and chemical fixation methods to identify a unique polymerization form of actin, both in the nucleus and in the cytoplasm. To prove that they really saw actin cables, authors labelled them with anti-actin antibodies and visualized by nano-gold; additionally, some cells were subjected to the treatment with an actin-depolymerizing drug – latrunculin, before fixation. After that, no cables remained in the cells.

On the whole, the paper is not very convincing. Images of the high pressure frozen yeast cells are acceptable, but the interpretation is not good enough. Below, the Major Concerns are listed:

1) The title of the paper is incorrect. It may be used as a running title, but the authors need to find more appropriate name for the primary title.

>> To be accurate in what we have found, we changed the title as follows "Nuclear bundles/cables containing actin in yeast meiosis ". As pointed out, together with the preprint by Ma et al., we soften our claim and avoid misunderstanding that nuclear bundle described in the paper is a conventional actin cable.

2) The immuno-gold labelling showed some gold particles associated with a bunch of the cables in the cytoplasm (Fig. 2a), but some gold particles were also identified in the cytoplasm. The authors suggest that these might be filasomes; they mentioned that filasomes are a "novel actin-containing membrane-less sub-cellular structure", but the paper mentioning filasomes was published more than 15 years ago, which is not novel enough, in my opinion. Additionally, those filasomes from 2003 do not look similar to the current Fig. 2A. I suggest that this is just non-specific binding.

>> Since the description on filasome is not a main part of this paper and we need more careful evaluation on our immuno-gold labelling on filasome and cannot deny the possibility of background staining as pointed out. We soften our claim gold labelling on filasome (page10, second paragraph, last two lines). Moreover, we added more images for fair presentation of data of immune-gold labelling (Supplemental Figure 6). In cytoplasm, we see gold labels on near vesicle and electron dense regions free of ribosomes.

3) Line 325 – The authors speculate that actin cables may form a three-dimensional structure in the meiotic nuclei, but they did not attempt to prove this, for example, using the tomography approach.

>> Serial section (60nM) shown in Fig. 5 and 6 clearly showed three-dimensional distribution of nuclear cables. Moreover, as described above, the pre-printer paper (Ma et al, BiorRxiv) nicely showed a structure of filaments in the bundle by Cryo-ET, which is complimentary to our data.

4) The TEM images in Fig. 5 do not give an impression that the discovered actin cables have a novel branch pattern, schematized in Fig. 8. The serial sections in Fig. 6-7 are too thick to accommodate a single actin cable. Why should the "branches" be attached to the main cable? Maybe they just cross the plane with the main cable? Again, this issue may be resolved by the tomography approach.

>> We agreed with this reviewer's point. At this point, the description of branches is inappropriate without further analyses on the structure. We removed branched nature of the cables/bundles throughout the most part of the text. When we mentioned "branched" structure, we rather used "branch-like" (page 7, first paragraph).

5) Line 294 - The authors declare that they find an alpha-actinin-like protein in

the budding east, but this suggestion is based only on the distance between filaments in the bundle. No immunoblotting was done to prove this.

>> We agreed with this reviewer's point. We removed our sentences about alpha-actinin-like protein, since we did not demonstrate its presence. Moreover, as mentioned, the nuclear bundles are unlikely to contain a typical actin filament.

Minor issues:

Fig 3e - Angels should be angles

>> We changed as pointed out.

REVIEWERS' COMMENTS:

Reviewer #1 (Remarks to the Author):

Since the first submission of this work by Takagi and collaborators, in late 2019, the authors have made substantial improvements to their work. They have added new data (Spo11 mutant), improved immuno-gold labeling and amended their text so as not to over-interpret their results. In particular authors have renamed "actin cable" into "bundles containing actin" the structures reported in this ms. The recent submission in BioRxiv (to be clearly cited in the references) of similar structures further validates the submitted work.

I nevertheless recommend to work on the text in English. Some parts like the last paragraph "Nuclear and cytoplasmic bundles contain actin" lack clarity. The title could still be improved.

Reviewer #2 (Remarks to the Author):

In this study, Takagi et al. describe the presence of structures with "feather-like" appearance in the cytoplasm and in the nucleus that they named as nuclear and cytoplasmic bundles. These structures are predominant during meiotic prophase (5h after induction), but a small percentage also appeared at an early time (2h after meiotic induction). They compare nuclear and cytoplasmic bundles and analyze different parameters as its diameter, average distance between bundles, length of bundles, and angle between the main bundle and branched structures. They use different electron microscopy techniques, including rapid freezing and chemical fixation, to demonstrate that these structures are not an artifact. To analyze the nature of these structures they use immuno-labeling with an anti-actin antibody and the presence of an actin-depolymerizing drug. Also, they demonstrate that in the spo11-Y135F mutant these structures are formed although the number is slightly reduced.

The presence of similar structures in the cytoplasm in response to different stress conditions has been reported in different organisms, but they had never been observed before in the nucleus of *Saccharomyces cerevisiae*. Although the nature of these structures and its possible function is not extensively explored in this manuscript, the discovery of these nuclear structures predominantly in meiotic prophase increases its novelty and interest to the field.

As Tagaki et al. mention in their manuscript, currently exists another manuscript posted on bioRxiv (<https://doi.org/10.1101/746982>) with similar and complementary results.

Major points

After the first revision, Tagaki et al. have toned down the conclusions about the nature of these structures, have added one of experiment that I had requested (spo11-Y135F catalytically inactive mutant), have improved text and some images of filosome, and have clarified some aspects, such as the manner to measure the average inter-filament distance. Despite these improvements, I would like to remark two main concerns remaining:

- 1) In the analysis of the dependence on actin polymerization (treatment with latrunculin B) (Lines 259-263) there is a high variability in the results obtained from two replicates with latrunculin B. In the first replicate, 8 of 14 nuclei (57%) display nuclear bundles, while in the second replicate only 3 of 21 nuclei (14%) are positive for nuclear bundles. Since this observation is critical for the conclusions reached, more cells should be analyzed to obtain more robust results.
- 2) Regarding the presence of actin in bundles, it is true that Tagaki et al. provide more images, but in these images some gold-particles also appear in other locations where the bundles are not appreciated, opening the possibility of non-specific binding of the antibody. I understand the difficulty of obtaining an accurate control for this experiment. Perhaps, a set of more EM images with anti-actin

antibody could be made available to be checked by the reviewers, and also a quantification of the percentage of bundles with actin associated/not associated should be done to make this experiment more convincing.

Minor points

1. The results of the posted bioRxiv manuscript, not yet accepted in peer review, should not be mentioned in the Results section. The sentences of page 6, lines 149-152 and page 9, line 265-267 should be moved to the Discussion section.
2. Please revised rawdata. Figures of excel pages are not accurately referenced and raw data of Supplementary Figure 5 is not attached.
3. Curiously, in the majority of images, nuclear bundles lie in a perpendicular position relative to microtubules. Perhaps, this disposition could be a clue for the discovery of the biological significance.
4. Page 5, line 142. The value of average distance is only one number, not a range.
5. Page 10, line 298. I'm confused with two sentences: "The distance of inter-filaments is mainly 10-15 nm" (page 10, line 298) and "An average distance between adjacent filaments is 8-15nm" (page 5, lines 141-142). Which one is correct?
6. Page 24, line 723. Change "envelop" to "envelope".
7. Page 24, line 748. Change "actin bundles" by "bundles".
8. Figure 1. Please, draw the ascus wall surrounding the four spores.
9. Figure 2, Add "e'" in the image.
10. Figure 4. The graph of "c" and "d" are interchanged: "c" is angles and "d" is length of cables. What is quantified in figure 4e and Supplemental Figure 5, the number of bundles or the number of bundles with actin? I think that the confusion is a consequence of the change in the name of the structures. If it is true, please change nuclear and cytoplasm actin by nuclear and cytoplasm bundles.
11. Figure 7. Please explain if "c" is a magnified image of "b". "d" is not observed in "b", does it correspond to another cell or section?
12. Figure 8. Explain if "b" is a magnified image of "a". Show complete cell of "c" and "d" to demonstrate that magnified images "c" and "d" correspond to cytoplasm and nucleus. In figure 8c, the identification of filosome is not very convincing if it is compared to supplemental figure 6. Quantification of ribosome density in this zone could be helpful to confirm that the structure shown in Figure 8c is really a filosome.
13. Supplementary figure 3. Is magnified image "e" part of the cell shown in image "c"? Please, clarify this issue.
14. Supplementary figure 6. It is a bit risky to mark the possible synaptonemal complex (SC)-like structure since it has not been previously observed with this technique in budding yeast. Could it be also an invagination of the nucleolus?. This image is really intriguing, but I would recommend adding a question mark to the SC labeling.
15. Supplementary figure 7. Do magnified images "d" and "e" correspond to the cell in image "a"?

Reviewer #3 (Remarks to the Author):

The revised manuscript Takagi et al. (COMMSBIO-19-1264A) describes the actin-like structure in the cytoplasm and nuclei of budding yeast, undergoing meiosis. After revision authors addressed all my questions:

1) Changed slightly the title;

2) Added more immuno-gold images;

3) They used serial sectioning to show the three-dimensional distribution of cables;

To address two other concerns, they just removed the corresponded mentioning from the text.

In this form, the manuscript may be accepted.

COMMSBIO-19-1264A

Responses to reviewers:

We do appreciate for constructive comments on our paper by the reviewers again, which are helpful for the improvement of our paper. We revised the paper based on the advice from the editor and reviewers. Our responses below are shown in green.

The editor:

< Please provide quantification of bundles observed with and without actin

-We added a quantified data on immuno-EM and mentioned the value in the main text (page 9, line 11-14).

< Please provide additional EM images in SI and/or deposit to Figshare/IDR

-We added additional EM images for immuno-gold EM in Supplemental Figure 8 and uploaded lots of additional EM images for each Figure in the FigShare (<https://doi.org/10.6084/m9.figshare.15049380>), which is not open to public at this point; we will open once our paper is published or are requested by the journal.

< Please provide information on how many times experiments were repeated with similar results in all figure legends.

-We added the information on reproducibility of each experiment (the number of the experiments) in Figure 4 and Supplemental Figure 2 and 5.

< Add the BioRxiv paper to the reference list, remove the citation from results and mention it as non-peer reviewed study only in the Discussion section.

-We moved the citation of the BioRxiv paper from Results to Discussion and added it in a reference list.

#1 reviewer:

I nevertheless recommend to work on the text in English. Some parts like the last paragraph "Nuclear and cytoplasmic bundles contain actin" lack clarity. The title could still be improved.

-We asked professional English editing service (Springer Nature Author Service: the certificate is attached to the end) after careful revision.

-We corrected some sentences including ones in the last paragraph to make clarity.

#2 reviewer:

Major points

1) In the analysis of the dependence on actin polymerization (treatment with latrunculin B) (Lines 259-263) there is a high variability in the results obtained from two replicates with latrunculin B. In the first replicate, 8 of 14 nuclei (57%) display nuclear bundles, while in the second replicate only 3 of 21 nuclei (14%) are positive for nuclear bundles. Since this observation is critical for the conclusions reached, more cells should be analyzed to obtain more robust results.

-As pointed out, we counted more cells for the time courses with LatB treatment by re-examination of EM samples, and new results are shown in page 9, line 11-14 in the first paragraph.

2) Regarding the presence of actin in bundles, it is true that Tagaki et al. provide more images, but in these images some gold-particles also appear in other locations where the bundles are not appreciated, opening the possibility of non-specific binding of the antibody. I understand the difficulty of obtaining an accurate control for this experiment. Perhaps, a set of more EM images with anti-actin antibody could be made available to be checked by the reviewers, and also a quantification of the percentage of bundles with actin associated/not associated should be done to make this experiment more convincing.

-We agree with this point. On the other hand, actin is very abundant, widely-distributed protein in both cytoplasm and nucleus as a monomer, polymers and a component of protein complexes. It is not surprising to us to see some gold particles in various regions in our sections. To respond this comment, we added more images of immuno-gold labelling in Supplemental Figure 8 and uploaded several images in Figshare (<https://doi.org/10.6084/m9.figshare.15049380>) for fair evaluation on our data sets.

Minor points

1. The results of the posted bioRxiv manuscript, not yet accepted in peer review, should not be mentioned in the Results section. The sentences of page 6, lines 149-152 and page 9, line 265-267 should be moved to the Discussion section.

-We cited results in the BioRxiv preprint only in Discussion and added it in the reference list.

2. Please revised rawdata. Figures of excel pages are not accurately referenced and raw data of Supplementary Figure 5 is not attached.

-We revised raw data in Excel file and added raw data in Supplemental Figure 5a-c.

3. Curiously, in the majority of images, nuclear bundles lie in a perpendicular position relative to microtubules. Perhaps, this disposition could be a clue for the discovery of the biological significance.

-Thanks. We pointed out this possible “spatial” tendency of nuclear bundles in the text (page 10, line 8-9 from the last paragraph).

4. Page 5, line 142. The value of average distance is only one number, not a range.

-We corrected this.

5. Page 10, line 298. I’m confused with two sentences: “The distance of inter-filaments is mainly 10-15 nm” (page 10, line 298) and “An average distance between adjacent filaments is 8-15nm” (page 5, lines 141-142). Which one is correct?

-Sorry for this confusion. As shown in Figure 4B, 8-15 nm is correct.

6. Page 24, line 723. Change “envelop” to “envelope”.

-We corrected it.

7. Page 24, line 748. Change “actin bundles” by “bundles”.

-We corrected it.

8. Figure 1. Please, draw the ascus wall surrounding the four spores.

-We corrected it.

9. Figure 2, Add “e’ ” in the image.

-We corrected it.

10. Figure 4. The graph of “c” and “d” are interchanged: “c” is angles and “d” is length of cables.

-We corrected it.

What is quantified in figure 4e and Supplemental Figure 5, the number of bundles or the number of bundles with actin?

-The number of bundles is correct. Fig. 4e shows a percentage of sections containing bundles in either nucleus or cytoplasm.

I think that the confusion is a consequence of the change in the name of the structures. If it is true, please change nuclear and cytoplasm actin by nuclear and cytoplasm bundles.

-Yes, we carefully checked and corrected by not using actin cables/bundles.

11. Figure 7. Please explain if “c” is a magnified image of “b”. “d” is not observed in “b”, does it correspond to another cell or section?

-We corrected it.

12. Figure 8. Explain if “b” is a magnified image of “a”. Show complete cell of “c” and “d” to demonstrate that magnified images “c” and “d” correspond to cytoplasm and nucleus.

-We corrected it. The image of the complete cell of c and d are out of focus, I will put it in Figshare instead of Fig.8.

In figure 8c, the identification of filosome is not very convincing if it is compared to supplemental figure 6. Quantification of ribosome density in this zone could be helpful to confirm that the structure shown in Figure 8c is really a filosome.

-We added density of ribosomes in filosome and cytoplasm as a control (legend in Supplemental Figure 6)-“An average density of ribosomes per a filosome is 3.3 ± 0.41 ribosomes pixel⁻² (n=13) while that in cytoplasm is 22.4 ± 0.96 (n=26). Bar indicates 100 nm.”

.

13. Supplementary figure 3. Is magnified image “e” part of the cell shown in image “c”? Please, clarify this issue.

-We corrected it.

14. Supplementary figure 6. It is a bit risky to mark the possible synaptonemal complex (SC)-like structure since it has not been previously observed with this technique in budding yeast. Could it be also an invagination of the nucleolus?. This image is really intriguing, but I would recommend adding a question mark to the SC labeling.

-Yes, this review is right. The structure might be related to nucleolus. We added “?” in the caption and figure of it..

15. Supplementary figure 7. Do magnified images “d” and “e” correspond to the cell in image “a”?

-We corrected it.

This document certifies that the manuscript
**Ultrastructural analysis in yeast reveals a meiosis-specific actin-containing nuclear
bundle**

prepared by the authors

Tomoko Takagi, Masako Osumi, Akira Shinohara

was edited for proper English language, grammar, punctuation, spelling, and overall style
by one or more of the highly qualified native English speaking editors at SNAS.

This certificate was issued on July 23, 2021 and may be verified
on the SNAS website using the verification code **A6F4-26E0-9DEB-395D-480D**.

Neither the research content nor the authors' intentions were altered in any way during the editing process. Documents receiving this certification should be English-ready for publication; however, the author has the ability to accept or reject our suggestions and changes. To verify the final SNAS edited version, please visit our verification page at secure.authorservices.springernature.com/certificate/verify.
If you have any questions or concerns about this edited document, please contact SNAS at support@as.springernature.com.

SNAS provides a range of editing, translation, and manuscript services for researchers and publishers around the world.
For more information about our company, services, and partner discounts, please visit authorservices.springernature.com.